# Beyond Staging: The Role of Pressure Ulcer Site and Multiplicity in Hospital Mortality and Length of Stay

**DOI:** 10.3390/healthcare13212815

**Published:** 2025-11-05

**Authors:** Dimitrios Zikos, Philip Eappen

**Affiliations:** 1Department of Healthcare Management and Leadership, Texas Tech University Health Sciences Center, Lubbock, TX 79430, USA; 2Shannon School of Business, Cape Breton University, 1250 Grand Lake Rd, Sydney, NS B1M 1A2, Canada; philip_eappen@cbu.ca

**Keywords:** patient safety, pressure ulcers, Medicare, length of stay, hospital mortality

## Abstract

**Highlights:**

**What are the main findings?**
The anatomical site and presence of multiple pressure ulcers were stronger predictors of prolonged hospital stay and increased mortality than ulcer stage alone.Certain ulcer sites, such as sacral, hip, head, buttock, and upper back, were independently associated with a higher risk of poor outcomes.

**What are the implications of the main finding?**
Risk assessment and prevention strategies should integrate ulcer site and multiplicity, not rely solely on staging frameworks.Systematic documentation, early detection, and patient safety protocols can reduce both clinical harm and the hospital resource burden of pressure ulcers.

**Abstract:**

**Background/Objectives:** Pressure ulcers are an important patient safety concern. While staging frameworks guide clinical management, the association between anatomical site, stage, and multiple PU presence and outcomes such as length of stay (LOS) and mortality in Medicare patients has not been fully characterized. The study objective is to examine the relationship between PU site, stage, and multiplicity and inpatient LOS and mortality among hospitalized Medicare patients. **Methods:** A cross-sectional study was conducted with 1,123,121 inpatient Medicare admissions from a 2019 Centers for Medicare and Medicaid Services (CMS) medical claims dataset. PUs were identified using ICD-10-CM codes classified by anatomical site and stage (1 through 4, unstageable, unspecified). Multiple regression models examined associations between PU characteristics and LOS and mortality, adjusting for age, sex, primary diagnosis, and hospital transfer. An analysis was conducted using SPSS version 29. **Results:** Stage 2 ulcers were the most common (28.6%), while unstageable or unspecified ulcers were frequent in the heels and head. The sacral region, buttocks, and heels were the most common anatomical sites. LOS gradually increased from Stage 1 (9.4 days) to Stage 4 (15.2 days). While the death rate did not increase consistently with stage, it was highest for upper back (14%), head (12.8%), and unspecified hip (12.8%) sites. Multiple regression was conducted to examine the association between PU locality, stage, and multiplicity and mortality (logistic regression) and LOS (linear regression). After controlling for patient demographics, admission, and clinical information, the regression results showed that the presence of multiple PUs, as well as anatomical sites of sacral, hip, head, buttock, and upper back ulcers, is associated with prolonged LOS and increased mortality. The presence of an advanced PU stage was found to be associated with prolonged LOS but not with inpatient mortality. **Conclusions:** The anatomical site and presence of multiple pressure ulcers were stronger predictors of prolonged hospital stay and increased mortality than ulcer stage alone. Certain ulcer sites, such as sacral, hip, head, buttock, and upper back, were independently associated with a higher risk of poor outcomes.

## 1. Introduction

Pressure ulcers (PUs) are a significant challenge for health systems, extending beyond the immediate concerns of wound management, and include broader patient safety issues [1]. PUs are injuries to the skin and underlying tissues that result from long mechanical loading and are more frequent in patients with limited mobility. The etiology of PUs includes patient factors, such as comorbidities, age-related tissue fragility, and nutritional deficiencies, as well as external factors, such as mechanical forces and hospital care environment characteristics [2,3,4,5,6,7]. The occurrence of PUs is both a clinical concern and a quality-of-care indicator, emphasizing the responsibility of health providers to prevent injury and to mitigate adverse outcomes [1,8].

Intensive care units (ICUs), long-term care acute settings, and nursing homes have higher PU incidence rates, likely due to patient acuity, hospital care practices, and issues with staffing, lack of educational initiatives, or prevention protocols [8,9]. Older age is a significant risk factor because of age-related skin changes, tissue integrity, reduced mobility, and the higher prevalence of comorbidities such as cardiovascular disease and diabetes [2,9]. Among younger populations, those mostly affected are patients with prolonged mechanical ventilation or extensive surgical interventions [4,9].

Prolonged pressure over bony prominences impairs tissue perfusion, leading to hypoxia, metabolic dysfunction, and, ultimately, cellular necrosis [2]. This is the reason patients with conditions such as compromised cardiovascular function and diabetes are particularly vulnerable [2,3]. In addition, hospital environmental factors play a critical role. Patients in critical care settings experience extended immobility, sedation, and exposure to medical devices, such as mechanical ventilators, catheters, and vascular access devices, all of which can contribute to tissue damage and delayed healing [4,5,6,10]. Staffing adequacy, nursing education, and institutional culture also influence PU outcomes, with insufficient knowledge of prevention protocols or a lack of systematic risk assessment contributing to higher incidence rates [7,11].

Evidence-based strategies for PU prevention include evidence-driven risk assessment, targeted interventions, and continuous monitoring. Validated instruments, such as the Braden Scale and Norton Scale, are available to measure risk and facilitate early identification of high-risk patients and focus on preventive resources at specific anatomical sites [12,13]. Repositioning protocols and the use of pressure-relieving surfaces reduce mechanical loading and forces, lowering the likelihood of tissue injury [14,15,16]. Positioning techniques (e.g., 30-degree lateral rotation method) can be particularly effective in reducing pressure over bony prominences [16]. Malnutrition also increases susceptibility to PUs (affecting tissue repair and skin integrity), and this emphasizes the need for nutritional assessment as part of PU prevention programs [17].

Health provider education has been shown to improve knowledge and attitudes about patient safety, which, in turn, contribute to safer clinical practices related to PU prevention [11,18]. Quality monitoring includes process measures (e.g., completion rates of risk assessments and preventive interventions) and outcome measures (e.g., PU incidence, stage, and severity) as well [19,20,21].

The clinical consequences of PUs extend beyond local tissue injury, affecting patient quality of life, functional status, and mental health. Patients report pain, sleep disturbances, mobility limitations, anxiety, depression, and reduced self-efficacy [22,23,24,25,26]. These impacts often influence long-term recovery and rehabilitation outcomes and extend beyond the hospital stay. Patients with advanced-stage PUs have a higher risk for infection, sepsis, and delayed recovery from underlying medical conditions, which have been associated with increased morbidity and mortality [27,28,29,30,31]. In addition, multiple co-existing PUs create compound challenges, reflecting underlying vulnerability, nutritional compromise, and complex care needs, further straining healthcare resources [2,32,33]. The psychological and social implications of multiple PUs, including body image, social withdrawal, and caregiver strain, emphasize the importance of patient-centered approaches that address physical and psychosocial dimensions of care [26,34].

PU staging provides a framework for assessing injury severity and includes four stages. While Stage I PUs can easily be managed [30,35], Stage II PUs require specialized wound care [36]. Stages III and IV PUs, though, are associated with increased risk of infection, sepsis, and prolonged recovery [31] and increase resource utilization [24].

The anatomical site and multiplicity of PUs also influence clinical outcomes. The sacrum, heels, buttocks, hips, and elbows are the most affected regions, and multiple co-existing ulcers indicate compounded clinical challenges [2,32]. Multiple PUs are frequently linked with malnutrition, advanced frailty, limited healing capacity, increased risk for infection, prolonged hospitalization, and increased mortality risk [2,32,33]. The psychological burden of multiple ulcers compounds patient vulnerability and interventions that address mental health alongside physical wound care [26,34].

While prevention strategies and risk assessment tools are well established, there remains a need to quantify the frequency, stage, and anatomical characteristics of PUs and evaluate their association with clinical outcomes. Mapping PU prevalence and PU characteristics can offer a useful tool to understand early risk detection, preventive interventions, and improved resource allocation. Moreover, understanding the relationship between PUs and adverse outcomes provides evidence to support quality improvement initiatives and patient safety policies. The present study aims to address this knowledge gap by (i) mapping the frequency, anatomical site, stage, and characteristics of PUs and (ii) examining their association with inpatient LOS and hospital mortality (all-cause) among hospitalized elderly patients. This study was not designed to explain the mechanism or clinical cause of comorbidities, but it focuses on their burden and the patterns of PUs that are associated with a higher burden for the two study outcomes. By combining descriptive analyses of PU patterns with outcome associations, this study aims to improve our understanding of PU burden in hospital settings. The findings could also guide prevention strategies, inform clinical risk stratification (and help healthcare providers prioritize high-risk patients), optimize care processes, and support evidence-based patient safety interventions.

## 2. Materials and Methods

### 2.1. Study Design

A retrospective cross-sectional study using secondary administrative data was designed to examine hospitalized Medicare patients with documented pressure ulcers. The first goal of this research was to create a mapping of the PUs, including their frequency, the most frequent combinations, and their stages. Goal 2 was to examine the association between PU site, stage, and multiple PU presence and two critical hospital outcomes: inpatient mortality and LOS. After the bivariate analysis was completed, this study furthermore controlled for demographics, primary Dx, and source of admission. Figure 1 shows a diagrammatic representation of the study design.

### 2.2. Dataset and Study Variables

This study used a dataset of 1,123,121 Medicare beneficiary inpatient admissions. The CMS Limited Data Set (LDS) Inpatient dataset is a de-identified claims dataset that includes information on inpatient hospital stays for Medicare patients for the year 2019. The dataset includes every Medicare inpatient admission that happened in hospitals located in the United States during that year. The dataset is made available directly via CMS. The researchers did not extract the original data themselves. Instead, the extracted and cleaned data is made available directly through CMS. CMS itself compiles the dataset and releases it on an annual and quarterly basis. It contains data on patient demographics, diagnoses (ICD-10-CM codes), procedures, LOS, discharge status, hospital charges, and other administrative details. Although patient identifiers are removed, the dataset has important variables that enable health services and outcomes research at a national level in the United States.

Several derived variables were created from the dataset, as appropriate, to examine how the PU characteristics of (i) locality, (ii) stage, and (iii) multiple PU presence are associated with the hospital mortality and LOS. All admissions were reviewed for the presence of PUs using ICD-10-CM Dx codes. Each unique PU ICD-10-CM code was extracted and coded as a dichotomous variable (present/absent) for analysis. A total of 25 groups of ICD-10-CM PU codes were identified. These codes included anatomical site-specific designations (e.g., sacral, heel, back) and were further detailed using secondary billable ICD-10-CM codes that captured the stage of the PU, classified as Stage 1, Stage 2, Stage 3, Stage 4, or unstageable. A summary of all extracted codes, along with their frequency of occurrence in the dataset, is provided in Table 1. The pressure ulcers were identified by scanning all ICD-10-CM medical diagnosis codes (primary and secondary) per patient, looking for codes representing pressure ulcers. PU codes were identified by the official CMS online tool (https://icd10cmtool.cdc.gov/?fy=FY2024, accessed on 3 November 2025) by using the keyword “pressure ulcer”. The codes were validated by the pressure ulcer code list from icd10Data.com. A total of 149 pressure ulcer codes were identified, representing 25 different anatomical sites. There are different codes for left and right anatomical sites (e.g., left elbow vs. right elbow PUs are represented by different codes). PUs on patients who had a PU on both sites of the same location were treated as different PUs and not merged. We made this decision because of the different implications that opposite sites (left vs. right) may have, such as underlying clinical and functional factors, instead of just being a random occurrence.

For each site, there are multiple child codes, each representing the pressure ulcer staging information. Readmission information was not available since the unit of analysis of the dataset is the hospital admission, and the patient id information is not available. We did not exclude any of the pressure ulcer codes from the analysis since the classification was well defined and unambiguous.

Comorbidities were not included as covariates because they are often interrelated with both ulcer development and outcomes. This could have obscured these primary relationships. In other words, comorbidities may function as mediators rather than pure confounders reflecting patient frailty that lies on the pathway between pressure ulcer severity and mortality or length of stay. We did decide to control for patient principal diagnosis because its presence allows for the adjustment for the clinical context in which PUs occurred (i.e., distinguishing surgical from medical cases).

In addition to analyzing individual PU anatomical sites, composite constructs representing the presence of two or more co-existing PU anatomical sites during the same hospitalization were developed. These constructs were created by identifying admissions in which multiple distinct ICD-10-CM PU codes (indicating different anatomical sites) were documented simultaneously. Dichotomous variables were generated to indicate the presence of multisite PUs, allowing for comparisons between patients with single and multisite PUs.

Therefore, with these data transformations, it was possible for the present study to examine three different PU characteristics: (i) PU anatomical site, (ii) PU stage, and (iii) multisite PU presence (more than 1 PU site code).

This approach enabled the evaluation of the cumulative burden of PUs on inpatient outcomes. The presence of multiple PU anatomical sites was analyzed as an independent variable in the regression models assessing two key outcomes: hospital LOS and inpatient mortality. By including these multisite constructs, this study aimed to determine whether patients with multiple PU anatomical sites experienced worse clinical outcomes compared to those with ulcers at a single site, after adjusting for covariates, as shown in Table 1, which presents all the variables that this study used from the CMS dataset, their operational definitions, and their role in this research.

### 2.3. Statistical Analysis

Initial descriptive statistics were calculated, including the mean LOS, mortality rate, and distribution of PU stages and anatomical sites. To examine the relationship between PU characteristics and inpatient outcomes, multivariable regression analyses were conducted. Specifically, linear regression models were used to evaluate the association between PU anatomical site and hospital LOS. A logistic regression model was used to assess the association between PU anatomical site and inpatient mortality. All regression models controlled for potential confounders, including primary Dx, age group, sex, and admission transfer from another setting/SNF.

Statistical analyses were performed using IBM SPSS Statistics for Windows, Version 29.0 (IBM Corp., Armonk, NY, USA). A two-tailed alpha level of 0.05 was used to determine statistical significance.

## 3. Results

### 3.1. Mapping of Pressure Ulcers

Out of 1,123,121 patient admissions, 41,525 patients (3.69%) had at least one documented PU. Among these, 29,460 patients (70.94%) had a PU at a single site, 7630 patients (18.37%) had two PUs, 2954 patients (7.11%) had three PUs, and 912 patients (2.19%) had four PUs. Only 569 patients (1.37%) had five or more PUs. In total, 60,877 unique PU codes were recorded in the dataset. Regarding ulcer stage distribution, 12.31% of the PUs were Stage 1, 28.64% Stage 2, 15.83% Stage 3, and 15.87% Stage 4. Additionally, 12.73% were classified as unstageable and 14.76% as unspecified. The distribution of stages varied significantly across anatomical sites. Table 2 highlights, for each PU site, the most frequent stage. For several sites, the “unstageable” or “unspecified stage” ICD-10-CM code was the more frequent code. As expected, PUs recorded at unspecified sites (e.g., unspecified elbow, heel, hip, or ankle) had the highest proportion of unstageable or unspecified cases. The right heel, left heel, and head followed, with nearly half of the cases at these sites lacking a definitive stage classification (See Table 2).

To summarize the severity of PUs by anatomical site, we calculated the weighted average stage for each site, excluding unstageable and unspecified cases. We call this metric the “Pressure Ulcer Locality–Stage score” (PULS).PULS = (1 × k1 + 2 × k2 + 3 × k3 + 4 × k4)/100
where kj represents the percentage of cases at stage j. The PULS score ranges from 1 (all cases Stage 1) to 4 (all cases Stage 4). The last column in Table 2 presents the PULS values across anatomical sites. With respect to the PULS score, the highest mean PU stage values were observed for contiguous site PUs (3.09), followed by left hip (3.07) and right hip (3.02). In contrast, the lowest scores were found for the head, left heel, and right heel (Table 2).

### 3.2. Bivariate Associations Between PU Properties and Study Outcomes

#### 3.2.1. Pressure Ulcer Site vs. Study Outcomes

The distribution of PU sites, mean LOS, and mortality rates is presented in Table 3. The sacral region was the most frequent site, with 23.5 cases per 1000 hospital admissions, followed by the right buttock (4.3 per 1000) and left buttock (4.2 per 1000). The mean LOS varied by site, with the longest hospitalizations observed for head PUs (16.22 days), left upper back PUs (16.03 days), and contiguous site PUs involving the back, buttock, or hip (15.26 days). The mortality rates were highest for left upper back PUs (14.04%), head PUs (12.82%), and unspecified hip PUs (12.80%).

#### 3.2.2. Pressure Ulcer Stage vs. Study Outcomes

Without considering the PU site, only the stage, those with any PU of Stage 1 had a mean LOS of 9.4 days (±11.15), which increased to 10.33 days (±12.09) for Stage 2 PUs, up to 12.88 days (±43.40) for Stage 3, and 15.24 days for Stage 4 PUs. For inpatient mortality, those with any PU of Stage 1 had a mortality rate of 7.12%, which increased to 7.91% for Stage 2 PUs and 8.66% for Stage 3. Interestingly, Stage 4 PU patients did not have a higher mortality rate than Stage 3 PU patients (8.62%).

For most anatomical sites, advancing PU stage did not correspond to higher mortality. Simple linear regression analyses indicated a moderate to strong positive trend for sacral PUs (R^2^ = 0.882, *p* = 0.06) and PUs of unspecified sites (R^2^ = 0.896, *p* = 0.05). Conversely, several PU sites demonstrated an inverse association, with higher-stage ulcers linked to lower mortality (Table 3).

In contrast, LOS exhibited a more consistent pattern. For most PU sites, LOS increased progressively from Stage 1 through Stage 4. Notably, sacral PUs (R^2^ = 0.96, *p* = 0.01) and left hip PUs (R^2^ = 0.99, *p* < 0.01) demonstrated near-perfect linear relationships between ulcer stage and hospital LOS. The results for all PU sites are summarized in Table 4.

#### 3.2.3. Multiple-Site Pressure Ulcers vs. Study Outcomes

A significant increase in both LOS and hospital mortality was observed among patients with PUs at multiple anatomical sites. The correlation between LOS and the total number of PU codes was statistically significant (Pearson’s r = 0.088, *p* < 0.01). An independent samples *t*-test demonstrated a significant difference in the mean number of PU codes between patients who died in the hospital and those discharged alive (t = −56.127, *p* < 0.001). The mean LOS increased progressively with the number of PU sites: 10.58 days for patients with one PU, 11.63 days for two, 12.34 days for three, and 23.46 days for eight or more. A similar trend was observed for mortality, with death rates of 8.13% for one PU, 8.24% for two, 9.51% for three, and 12.50% for eight or more (Figure 2).

As Table 5 shows, the top three most frequent combinations of multiple co-existing PU sites were found to be {right heel, left heel}, {sacral, left heel}, and {sacral, right heel}. Some multiple-PU–site combinations were found to be associated with especially prolonged hospital stays and high hospital mortality rates. Table 5 presents the top 10 combinations of PU sites with the highest hospital inpatient mortality rate and LOS for combinations that appeared in at least 1 in 10,000 cases.

### 3.3. Multivariate Analysis

#### 3.3.1. Length of Stay

To assess the association between PU anatomical site and hospital LOS, a multiple linear regression analysis was performed. Twenty-five dichotomous variables representing distinct PU sites were entered into the model using the stepwise elimination method. The total number of different PU Dxs and the PU staging information were also inserted into the model. The control variables included primary Dx (categorized using Clinical Classifications Software [CCS]), patient transfer status (from a different unit or another hospital or a SNF), sex, and age group.

Several PU anatomical sites were significantly associated with hospital LOS, including the total number of PUs, the presence of head, sacral, right hip, right buttock, left buttock, left upper back, and contiguous site PUs (Table 6).

The presence of a PU of a stage other than Stage 1 was found to be associated with prolonged hospital stay. Table 6 shows the results of the multiple linear regression.

#### 3.3.2. Inpatient Mortality

A Binary Logistic Regression was used to examine the association between PU site, stage, and complexity and inpatient mortality. After controlling for clinical, demographic, and admission factors, several PU sites continued to be associated with an increased likelihood for hospital death, as shown in Table 7, with the most strongly associated being the left upper back PU, sacral, left buttock, right heel, right hip, and head (OR = 1.64, 1.07–2.52).

Of the control variables, age and transfer from another hospital or SNF were also found to be associated with an increased likelihood for inpatient death, while PU stage was not found to be a predictor of inpatient mortality. Table 7 presents the statistically significant variables.

## 4. Discussion

This study analyzed 1.1 million Medicare inpatient admissions to map the characteristics of PUs and examine their association with LOS and inpatient mortality. The overall PU prevalence in this hospitalized Medicare population was 3.7%, which is consistent with national estimates. The most common anatomical sites were the sacral region, buttocks, and heels, likely due to immobility-related pressure. Stage 2 ulcers were the most frequently documented stage. However, a significant portion of ulcers, particularly on the heels and head, were classified as unstageable or unspecified.

PU severity, as measured by our Pressure Ulcer Locality–Stage score, was highest in the hips and contiguous back/buttock/hip sites and lowest in the heels and head. Unstageable or unspecified ulcers were most frequent in the hips, heels, ankles, and elbows. The bivariate analysis also highlighted the following sites with the highest mortality: left upper back (14.04%), head (12.82%), and unspecified hip (12.80%).

The central finding was that PU anatomical site and multiplicity were stronger predictors of LOS and mortality than PU stage alone. Multiple regression showed that PUs on the sacral region, hip, head, buttock, and upper back were risk factors for both prolonged LOS and increased inpatient mortality. LOS and mortality also increased progressively as the number of concurrent PUs rose. For example, patients with eight or more ulcers had a mean LOS of 23.5 days and a mortality rate of 12.5%, compared to 10.6 days and 8.1% for those with just one ulcer. Conversely, while LOS increased consistently with advancing stage (from 9.4 days for Stage 1 to 15.2 days for Stage 4), higher PU stage was not a significant predictor of inpatient mortality.

The finding that site and multiplicity are more predictive than stage suggests that PUs should be viewed not just as a localized skin injury, but as a proxy for systemic vulnerability. The specific high-risk sites are associated with immobility and are likely indicators of systemic frailty. The presence of multiple ulcers shows a compounding effect where healing capacity is compromised. The location and number of PUs, therefore, appear to carry more prognostic information about a patient’s overall health status than the depth of a single wound [2].

We believe that the high frequency of Stage 2 ulcers is an issue of detection and subsequent documentation. Stage 1 PUs are often subtle or underreported in busy clinical settings. Stage 2, on the other hand, comes with a clear break in the skin, prompting easier (and likely more consistent) clinical recognition and coding.

It is concerning that the rate of “unstageable” or “unspecified” ulcers is very high, as this indicates issues in clinical assessment and documentation. For example, heel ulcers are often covered with eschar that prevents staging until debridement; head ulcers in elderly patients may present atypically. We would like to point out that unstageable ulcers themselves may hold clinical significance, as they could be proxies of overall patient vulnerability.

We will attempt to explain the lack of association between advanced PU stage and mortality. Firstly, patients with the most severe underlying illnesses may not survive long enough to develop advanced-stage (Stages 3 or 4) ulcers. Second, hospitals that are more diligent in documenting and coding higher-stage PUs may also be those with more robust prevention and care protocols, which, in turn, reduce the impact of an ulcer on mortality. Finally, as noted in our methods, comorbidities were treated as mediators rather than confounders and were not included in the regression models. This means the mortality risk is likely driven by the underlying illness severity rather than the ulcer itself.

The strong association between PUs and LOS represents a “feedback loop”. Prolonged hospitalization increases the risk of developing PUs; in turn, the presence of a PU, particularly an advanced-stage or multiple PUs, increases infection risk, therefore extending the hospital stay [28].

The primary strength of this study is its large, national dataset, with high statistical power and generalizability to Medicare patients. However, we can identify some limitations. First, its reliance on ICD-10-CM codes from administrative data is subject to documentation and coding variability (i.e., a high number of “unstageable” ulcers). With the dataset, it is not possible to distinguish between pre-existing PUs and those acquired during hospitalization. We did use, though, the source of admission as a control variable to partly mitigate for this issue. We are aware that while this study identifies associations, it was not designed to establish causality.

Our findings have significant implications for clinical practice and patient safety. The evidence that anatomical site and multiplicity are stronger predictors of poor outcomes than stage alone can be the basis for rethinking risk assessment. Existing frameworks (such as the Braden and Norton Scales) should recognize specific PU locations and the presence of multiple ulcers as “red flag” indicators. This includes recognizing high-risk combinations, such as bilateral heels with sacral or hip involvement, which were associated with especially poor outcomes [37].

The high prevalence of unstageable or unspecified pinpoints the need for improved documentation and administrative investment in staff education and training on staging frameworks. We strongly believe that accurate classification is required for appropriate treatment, enabling the early detection of high-risk patients.

Our findings should trigger discussions to improve bedside PU prevention and management. We also recognize the need for better allocation for support devices as well as specific protocols for patients with ulcers in high-risk locations (sacral, hip, head, upper back) or multiple PUs. In addition, this translates to prioritized skin assessments for these patients. Adjustments to treatment pathways should also be considered to initiate earlier wound care for PUs at high-risk sites [38].

We believe that PUs should be viewed as both contributors and indicators of negative hospital outcomes. Prioritizing early detection, systematic prevention, and accurate documentation is critical to reducing both clinical harm to patients and the resource burden. Future research should focus on these high-risk PU patterns to understand specific patient subgroups and comorbidity constructs, which were beyond the scope of the present study. Future work should also examine organizational and structural hospital-level characteristics to understand how staffing, policies, technologies, ownership, or training gaps may contribute to the documentation inconsistencies that we found in this study.

## 5. Conclusions

The numerous unstageable ulcers, particularly in the heels and head, show ongoing challenges in documentation and staging accuracy. These findings suggest that training, resource availability, and assessment protocols can improve the classification and detection of PUs. PU site and multiplicity were stronger predictors of LOS and mortality than stage alone. The presence of multiple ulcers amplified these risks, supporting the interpretation that extensive ulceration signals a global decline in patient resilience.

A higher ulcer stage was not directly linked to increased mortality. This may be because of underlying illness severity or institutional coding and care practices that affect outcome patterns.

Overall, we believe that PUs should be viewed both as indicators and contributors to negative hospital outcomes. The feedback loop between prolonged LOS and PU severity shows the importance of early prevention, monitoring, and standardization in reporting. Validated risk assessments, pressure redistribution intervention, and staff education are foundational to controlling PU progression. Training should focus on early recognition of initial and hard-to-recognize skin changes. These preventive measures will also lessen the effects of PUs on hospital resource use and system-level outcomes.

While this study was not designed to examine associations between PUs and comorbidities, future research should also explore hospital-level determinants and patient-level comorbidities to better identify high-risk patient subgroups.

## Figures and Tables

**Figure 1 healthcare-13-02815-f001:**
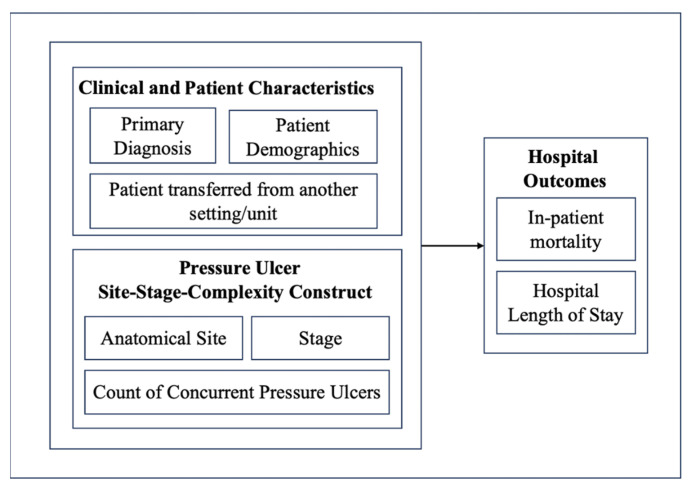
Diagrammatic Representation of the Study Design.

**Figure 2 healthcare-13-02815-f002:**
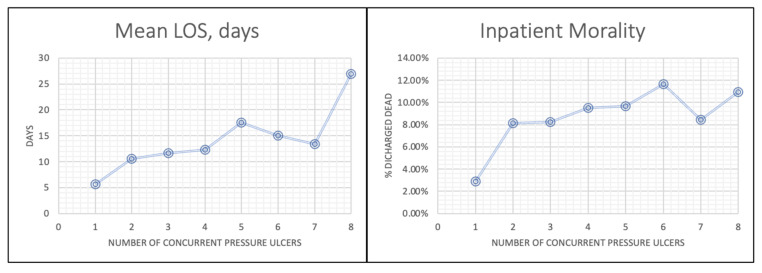
Mean LOS and Inpatient Mortality vs. Number of Concurrent PU Codes.

**Table 1 healthcare-13-02815-t001:** Variables Used in this Study and Operational Definitions.

Variable Name	How It Is Calculated	Definition	Values
Dependent Variables			
Length of stay (days)	Derived, calculated as the difference in discharge–admission date	Total number of inpatient days	Integer; length of stay (days).
Died	Derived from “discharge status”	Patient expired during hospitalization	Yes/no
Main Independent Variables			
Presence of PUs by site	Derived from secondary Dx list	The anatomical site where the PU is present, using ICD-10	One y/n variable per site/stage combo
Pressure ulcer stage	Derived, PU code of highest stage; for >1 PUs, it takes the value of the worst stage	PU stage, as recognized by health providers	1, 2, 3, 4
Number of concurrent PUs	Count of unique PUs regardless of stage/site	Sum of unique PU codes during the hospital stay	Integer; sum of total PU codes
Control Variables			
Age group	As it appears in the original dataset	Patient age based on the date of birth	5-year-range groups
Patient sex	As it appears in the original dataset	Administrative sex in the admission file	Male/female
Primary Dx	Categorical variable of the primary Dx	The patient Dx that led to need for hospitalization	285 CCS codes
Patient transferred from…			
Another hospital	Derived from “Admission Source”	Patient transferred directly from another hospital	Yes/no
Skilled nursing facility		Patient transferred directly from SNF	Yes/no
Another unit (same hospital)		Internal transfer from another unit of the hospital	Yes/no

**Table 2 healthcare-13-02815-t002:** PU Frequency by Stage and Pressure Ulcer Locality–Stage (PULS) Score per Anatomical Site.

	Stage 1	Stage 2	Stage 3	Stage 4	Unstageable/Unspecified	PULS
Unspec. Elbow	16.67% (3)	16.67% (3) *	5.55% (1)	0% (0)	61.07% (11)	1.71
Right Elbow	15.34% (27)	25.57% (45) *	19.32% (34)	9.09% (16)	30.68% (54)	2.32
Left Elbow	15.89% (31)	23.59% (46) *	16.41% (32)	14.35% (28)	29.73% (58)	2.42
Unspecified Back	17.91% (141)	32.4% (255) *	13.6% (107)	6.73% (53)	29.34% (231)	2.13
Right Upper Back	13.29% (25)	29.79% (56) *	23.4% (44)	9.57% (18)	23.92% (45)	2.38
Left Upper Back	16% (28)	24.57% (43) *	17.71% (31)	9.71% (17)	31.99% (56)	2.31
Right Low. Back	12.22% (11)	26.67% (24) *	23.33% (21)	13.33% (12)	24.44% (22)	2.50
Left Lower Back	4.76% (4)	28.57% (24) *	20.24% (17)	16.66% (14)	29.75% (25)	2.69
Sacral	12.48% (3363)	31.55% (8499) *	16.45% (4432)	20.18% (5437)	19.33% (5210)	2.55
Unspecified Hip	6.34% (8)	11.9% (15)	11.11% (14)	12.69% (16) *	57.93% (73)	2.72
Right Hip	6.23% (108)	17.96% (311)	18.19% (315)	32.39% (561) *	25.31% (437)	3.03
Left Hip	6.01% (101)	17.87% (300)	17.33% (291)	34.96% (587) *	23.81% (400)	3.07
Unspec. Buttock	19.27% (328)	34.67% (590) *	10.28% (175)	5.99% (102)	29.78% (507)	2.04
Right Buttock	10.71% (529)	41.37% (2042) *	15.94% (787)	12.62% (623)	19.33% (955)	2.38
Left Buttock	11.01% (526)	40.47% (1934)	16.38% (783)	12.15% (581)	19.97% (955)	2.37
Contiguous Site	4.51% (6)	20.30% (27)	18.79% (25)	36.84% (49) *	19.54% (26)	3.09
Unspec. Ankle	9.58% (7)	17.81% (13) *	8.22% (6)	8.21% (6)	56.15% (41)	2.34
Right Ankle	9.04% (69)	20.84% (159) *	19.4% (148)	12.18% (93)	38.52% (294)	2.57
Left Ankle	9.74% (72)	19.89% (147) *	19.22% (142)	9.6% (71)	41.54% (307)	2.49
Unspecified Heel	10.93% (28)	15.63% (40) *	7.81% (20)	4.29% (11)	61.31% (157)	2.14
Right Heel	16.42% (729)	15.23% (676) *	12.89% (572)	7.61% (338)	47.83% (2123)	2.22
Left Heel	16.30% (780) *	15.49% (741)	13.4% (641)	7.04% (337)	47.76% (2285)	2.21
Head	15.16% (42)	20.22% (56) *	13.36% (37)	6.13% (17)	45.11% (125)	2.19
Other Site	8.60% (398)	24.62% (1139) *	18.91% (875)	10.74% (497)	37.10% (1716)	2.51
Unspecified Site	11.26% (133)	21.68% (256) *	7.45% (88)	6.68% (79)	52.91% (625)	2.20

* pressure ulcer stage with the highest proportion of patients, per anatomical site.

**Table 3 healthcare-13-02815-t003:** Frequency of All-Stage PU Localities and their Mean LOS and Mortality.

PU Anatomical Site (Any Stage)	N	Cass per 1000 Admissions	LOS (Days)	% Died
Unspecified Elbow	18	0.016	14.89	11.11%
Right Elbow	175	0.156	11.05	8.52%
Left Elbow	195	0.173	11.2	9.23%
Unspecified Back	779	0.693	10.46	11.42%
Right Upper Back	185	0.164	13.04	8.11%
Left Upper Back	171	0.152	16.03	14.04%
Right Lower Back	90	0.080	9.99	4.44%
Left Lower Back	83	0.074	10.69	7.23%
Sacral	26,476	23.537	11.8	9.40%
Unspecified Hip	125	0.111	10.84	12.80%
Right Hip	1715	1.525	14.85	8.86%
Left Hip	1666	1.481	12.33	8.58%
Unspecified Buttock	1693	1.505	9.48	8.74%
Right Buttock	4900	4.356	12.29	6.98%
Left Buttock	4743	4.216	12.33	7.57%
Contiguous Site	133	0.118	15.26	8.27%
Unspecified Ankle	73	0.065	8.28	6.85%
Right Ankle	760	0.676	11.27	8.16%
Left Ankle	735	0.653	12.72	8.30%
Unspecified Heel	256	0.228	10.54	6.64%
Right Heel	4397	3.909	11.61	7.82%
Left Heel	4752	4.224	11.29	7.18%
Head	273	0.243	16.22	12.82%
Other Site	4367	3.882	11.88	8.17%
Unspecified Site	1172	1.042	7.99	9.22%

**Table 4 healthcare-13-02815-t004:** Hospital Mortality and Length of Stay for PUs of Different Stages.

	Inpatient Mortality				R^2^ (Sig.)	Length of Stay (Days)				R^2^ (Sig.)
	PU Stage					PU Stage				
	1	2	3	4		1	2	3	4	
Unspecified Elbow	33.3%	0%	0%	NA		7	24	8	NA	NA
Right Elbow	3.70%	11.11%	8.82%	6.25%	0.04 (0.78)	11.15	9.91	15.62	11.31	0.10 (0.67)
Left Elbow	12.90%	6.52%	6.25%	10.71%	0.07 (0.72)	9	9.65	13.03	13	0.85 (0.07)
Unspecified Back	9.22%	11.76%	9.35%	9.43%	0.03 (0.81)	9.21	9.96	9.78	13.36	0.70 (0.15)
Right Upper Back	12.00%	7.14%	9.09%	5.56%	0.65 (0.19)	5.4	10.38	20.3	19.56	0.87 (0.06)
Left Upper Back	7.14%	16.28%	6.45%	23.53%	0.39 (0.37)	12.96	12.5	19.16	34.19	0.80 (0.10)
Right Lower Back	9.09%	8.33%	0.00%	0.00%	0.83 (0.08)	7.55	9.75	10.48	10.58	0.80 (0.10)
Left Lower Back	25.00%	4.17%	11.76%	0.00%	0.62 (0.20)	11.5	7.58	16.88	11.71	0.11 (0.66)
Sacral	7.61%	8.95%	9.18%	9.67%	0.88 (0.06)	8.92	10.57	12.49	15.97	0.96 (0.01)
Unspecified Hip	25.00%	20.00%	7.14%	12.50%	0.67 (0.17)	6.5	5.53	11.29	13.27	0.81 (0.09)
Right Hip	7.41%	9.32%	8.89%	7.84%	0.01 (0.87)	9.33	10.02	13.89	22.1	0.86 (0.07)
Left Hip	5.94%	9.00%	9.62%	7.33%	0.13 (0.62)	7.83	9.78	12.46	14.71	0.99 (<0.01)
Unspecified Buttock	4.88%	8.47%	6.86%	9.80%	0.64 (0.19)	9.17	9.04	11.1	14.02	0.85 (0.07)
Right Buttock	5.67%	6.02%	8.39%	4.98%	0.001 (0.97)	8.79	10.01	18.66	16.84	0.74 (0.13)
Left Buttock	6.84%	6.72%	9.07%	5.16%	0.04 (0.78)	8.24	10.25	18.84	16.2	0.71 (0.15)
Contiguous Site	0.00%	7.41%	24.00%	4.08%	0.12 (0.64)	5.5	11.89	14.44	13.14	0.68 (0.17)
Unspecified Ankle	14.29%	7.69%	16.67%	0.00%	0.34 (0.41)	4.57	7.58	7.5	9	0.84 (0.08)
Right Ankle	11.59%	7.55%	8.78%	4.30%	0.77 (0.11)	9.57	10.52	13.89	12	0.53 (0.26)
Left Ankle	6.94%	4.08%	9.86%	9.86%	0.45 (0.32)	11.65	11.96	14.12	14.14	0.85 (0.07)
Unspecified Heel	3.57%	10.00%	5.00%	9.09%	0.22 (0.52)	13.71	8.03	12.25	11.91	0.01 (0.93)
Right Heel	7.13%	7.40%	6.64%	7.69%	0.07 (0.73)	11.65	10.34	13.36	13.82	0.58 (0.23)
Left Heel	8.08%	5.80%	4.99%	6.82%	0.19 (0.55)	10.65	10.63	11.41	14.06	0.76 (0.12)
Head	9.52%	8.93%	2.70%	5.88%	0.49 (0.29)	13.02	12.14	17.76	27.18	0.81 (0.09)
Other Site	8.79%	7.11%	8.34%	9.46%	0.17 (0.57)	11.08	10.17	14.04	14.39	0.71 (0.15)
Unspecified Site	3.01%	7.42%	7.95%	10.13%	0.89 (0.05)	7.29	7.75	9.1	9.97	0.97 (0.01)

**Table 5 healthcare-13-02815-t005:** PU Combinations with the Highest Inpatient Mortality and Longest Hospital Stay.

PU Combination	N	LOS (Days)	PU Combination	N	% Died
{Right Hip, Left Buttock, Sacral}	104	55.94	{Unspecified Buttock, Other Site}	108	18.52%
{Right and Left Buttocks, Right Hip}	101	55.69	{Unspecified Back, Sacral}	281	14.59%
{Right Hip, Right Buttock, Sacral}	115	52.76	{Right and Left Heels, Right Hip}	102	13.73%
{Right Hip, Left Buttock}	176	38.77	{Right Buttock, Other Site, Sacral}	202	13.37%
{Right Hip, Right Buttock}	209	35.11	{Right Ankle, Other Site, Sacral}	100	13.00%
{Right Buttock, Other Site, Sacral}	202	19.55	{Sacral, Right Heel}	146	12.90%
{Right and Left Buttocks, Sacral}	806	19.19	{Right and Left Buttocks, Right Hip}	101	12.87%
{Left Buttock, Other Site, Sacral}	215	18.56	{Right and Left Ankles}	159	12.58%
{Right Hip, Sacral}	904	18.53	{Left Elbow, Sacral}	104	12.50%
{Left Hip, Left Buttock, Sacral}	103	18.41	{Right and Left Heels, Other Site}	280	12.50%

**Table 6 healthcare-13-02815-t006:** Multiple Linear Regression to Examine the Association Between PU Site, PU Stage, and LOS.

Variable	b	SE	t	*p*-Value	95% CI
Total number of PU codes	0.933	0.077	12.073	<0.001	0.78–1.08
PU Stage					
Stage 1	−0.609	0.165	−3.695	<0.001	−0.93–−0.28
Stage 2	0.291	0.126	2.31	0.021	0.04–0.53
Stage 3	2.121	0.152	13.963	<0.001	1.82–2.41
Stage 4	4.099	0.165	24.907	<0.001	3.77–4.42
PU Site					
Sacral	2.189	0.116	18.925	<0.001	1.96–2.41
Left Hip	−2.146	0.308	−6.958	<0.001	−2.75–−1.54
Right Buttock	0.922	0.198	4.654	<0.001	0.53–1.31
Right Hip	2.096	0.304	6.891	<0.001	1.50–2.69
Head	4.175	0.679	6.151	<0.001	2.84–5.50
Left Buttock	1.027	0.199	5.164	<0.001	0.63–1.41
Left Elbow	−2.108	0.822	−2.565	0.01	−3.71–−0.49
Left Upper Back	2.889	0.862	3.351	<0.001	1.19–4.57
Right Lower Back	−2.996	1.171	−2.558	0.011	−5.29–−0.70
Right Elbow	−2.06	0.863	−2.388	0.017	−3.75–−0.37
Contiguous Site (Back/Buttock/Hip)	2.081	0.963	2.16	0.031	0.19–3.96
Right Ankle	−1.343	0.421	−3.191	0.001	−2.16–−0.51
Control variables					
Age Group	−0.058	0.006	−9.305	<0.001	−0.07–−0.04
Female Sex	−0.082	0.021	−3.83	<0.001	−0.12–−0.04
Transfer from Another Hospital	4.501	0.038	117.097	<0.001	4.42–4.57
Transfer from SNF	0.643	0.069	9.335	<0.001	0.50–0.77
Transfer from Same Hospital	2.375	0.089	26.694	<0.001	2.20–2.54
(Constant)	4.386	0.039	113.506	<0.001	4.31–4.46

**Table 7 healthcare-13-02815-t007:** Binary Logistic Regression analysis.

Variable Name	B	S.E.	Sig.	OR	95% CI OR
PU Stage					
Stage 1	−0.258	0.060	<0.001	0.772	0.68–0.86
Stage 2	−0.070	0.044	0.115	0.933	0.85–1.01
Stage 3	−0.087	0.052	0.092	0.916	0.82–1.01
Stage 4	−0.210	0.055	<0.001	0.811	0.72–0.90
PU Anatomical Site					
Left Upper Back	0.84	0.27	<0.001	2.32	1.38–3.89
Sacral	0.78	0.1	<0.001	2.19	1.79–2.68
Unspecified Hip	0.67	0.31	0.03	1.95	1.06–3.58
Right Hip	0.28	0.14	0.05	1.32	1–1.75
Unspecified Buttock	0.62	0.14	<0.001	1.86	1.42–2.42
Left Buttock	0.33	0.11	<0.001	1.39	1.12–1.73
Right Heel	0.3	0.12	0.01	1.36	1.07–1.72
Head	0.5	0.22	0.02	1.64	1.07–2.52
Other Site	0.39	0.12	<0.001	1.47	1.16–1.86
Unspecified Site	0.72	0.15	<0.001	2.05	1.54–2.74
Control Variables					
Age Group	0.156	0.003	<0.001	1.168	1.16–1.17
Female Sex	−0.135	0.011	<0.001	0.874	0.85–0.89
Transferred from Another Hospital	0.591	0.017	<0.001	1.806	1.74–1.86
Transferred from SNF	0.472	0.026	<0.001	1.603	1.52–1.68
Transferred from Same Hospital	−0.130	0.065	0.046	0.878	0.77–0.99
Constant	−4.09	0.02	<0.001	0.02	

## Data Availability

Restrictions apply to the external sharing of these data. The data were obtained from CMS after a data use agreement, which does not allow sharing; the data were only to be used for the purpose of the studies conducted by the researchers. The data can be obtained from CMS via a request process at https://www.cms.gov/data-research/cms-data/data-disclosures-and-data-use-agreements-duas (accessed on 3 November 2025).

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
