# Peer review of "Beyond Staging: The Role of Pressure Ulcer Site and Multiplicity in Hospital Mortality and Length of Stay"

_healthcare, 2025, doi:10.3390/healthcare13212815_

Round 1

Reviewer 1 Report

Comments and Suggestions for Authors

Dear authors,

I read with interest your paper about associations between anatomical site, stage, and multiplicity of pressure ulcers and hospital outcomes in Medicare inpatients. The study is original, timely, and clinically relevant for patient safety and resource optimization. However, the manuscript needs substantial methodological and editorial refinement to improve clarity and reproducibility.

The abstract reads more like a results summary and omits essential methodological details (such as the dataset year range, regression model structure, covariates, and statistical software). The conclusions should reflect the key findings without repeating information from the background.

The introduction is too long and includes pathophysiological details (mechanical load, tissue perfusion) that could be shortened. Moreover, the paragraph describing pressure ulcers (lines 108-116) is superfluous and redundant and can be removed. Citations are not always the most recent or directly relevant: I suggest updating the references with more current data.

In the Methods section, you present a large dataset (CMS), clear variable extraction procedures (ICD-10-CM), and well-structured tables. However, the timeframe of CMS data extraction is missing: please specify the year/s analyzed. I suggest clearly explaining how pressure ulcers were identified (e.g., primary vs. secondary diagnoses, exclusion criteria, and handling of readmissions). The use of “worst stage” and “multisite” constructs appears reasonable but requires validation. How was overlap between left and right sides managed? Adjusting only for age, sex, primary diagnosis, and transfer source seems insufficient; including comorbidity measures (e.g., Charlson Comorbidity Index) would strengthen the model. Finally, I recommend specifying the statistical software, significance thresholds, and confidence intervals consistently.

In the Results, the text frequently repeats data already presented in the tables. I suggest you to condense descriptions and highlight key patterns only. The PULS formula is useful, but rationale for weighting by stage severity should be justified and referenced. At present, no PULS index is recognized in the literature for assessing the severity of pressure ulcers based on their anatomical location.

The Discussion section requires a clearer structure: it should begin with the key findings, followed by their interpretation, then address strengths and limitations, and conclude with the implications. In its current form, the discussion blends speculative reasoning with the results. It is also important to avoid redundancy, as several sentences repeat the same idea. Finally, additional references should be included to support the claims made in this section.

The conclusions are somewhat repetitive and should be refined in light of the discussion.

Author Response

Thank you for your review. After an extensive meeting with my co-author, we read your critique and are explaining here how we addressed your questions. We included in the resubmission portal of MDPI, a “track-changes” version of the manuscript, so that you can see the changes we made.

Reviewer comment: The abstract reads more like a results summary and omits essential methodological details (such as the dataset year range, regression model structure, covariates, and statistical software). The conclusions should reflect the key findings without repeating information from the background.

Authors response: We re-structured the abstract to include dataset year, regression model structure (predictors, and control variables), and types of regressions used. We also mentioned the statistical software in the Methods section of the abstract. We also re-wrote the Conclusions section of the abstract to directly reflect the findings, avoiding generalized ideas or results not directly associated with findings. The updated resubmission now includes the updates. We are providing a track-changes version of the document to make it easier to review changes to the abstract.

Reviewer comment: The introduction is too long and includes pathophysiological details (mechanical load, tissue perfusion) that could be shortened.

Authors response: We do agree that the introduction includes details that are not required and are beyond the scope of the study. We made significant reductions to the introduction section, and especially, per your recommendation, we minimized “book knowledge” details, such as pathophysiological properties etc. We did add an extra sentence towards the end of the discussion section (where we explain the aim), stating that the study was not designed to explain the mechanism or clinical cause of comorbidities, but focuses on their burden, and the patterns of PUs that are associated with a higher burden for the two study outcomes. The introduction section has been reduced by at least 200 words.

Reviewer comment: Moreover, the paragraph describing pressure ulcers (lines 108-116) is superfluous and redundant and can be removed.

Authors response: The paragraph describing ulcers stages was reduced to just two lines. We decided it is appropriate to keep a few sentences briefly mentioning staging, because staging is an important element of the study. See below:

PU staging provides a framework for assessing injury severity and includes four stages.,. While stage I PUs can easily be managed. Stage II require specialized wound care. Stage III and IV PUs, though, are associated with increased risk for infection, sepsis, and prolonged recovery and increase resource utilization.

Reviewer comment: Citations are not always the most recent or directly relevant: I suggest updating the references with more current data.

Authors response: We updated several citations with most recent and relevant ones. We focused on replacing citations that were older than 5-6 years and had similar more recent citable evidence. Please review the updated version of the tracked changes file. Now the vast majority of the citated work is recent, with only a few exceptions of more foundational and widely articles.

Reviewer comment: In the Methods section, you present a large dataset (CMS), clear variable extraction procedures (ICD-10-CM), and well-structured tables. However, the timeframe of CMS data extraction is missing: please specify the year/s analyzed.

Authors response: We updated the “Datasets and Study Variables section” with this new information:

“The study used a dataset of 1,123,121 Medicare beneficiary inpatient admissions. The CMS Limited Data Set (LDS) Inpatient dataset is a de-identified claims dataset that includes information on inpatient hospital stays for Medicare patients for the year 2019. The dataset includes every Medicare inpatient admission that happened in hospitals located in the United States during that year. The dataset is made available directly via CMS. The researchers did not extract the original data themselves. Instead, the extracted and cleaned data is made available directly through CMS. CMS themselves compiles the dataset and releases them on an annual and quarterly basis.”

Reviewer comment: I suggest clearly explaining how pressure ulcers were identified (e.g., primary vs. secondary diagnoses, exclusion criteria, and handling of readmissions).

Authors response: Thank you, indeed, this information should have been included. We added a new paragraph on the discussion section explaining in detail how pressure ulcers were identified. This is the added paragraph:

“The pressure ulcers were identifying by scanning all ICD-10-CM medical diagnosis codes (primary and secondary) per patient, looking for codes representing pressure ulcers. PU codes were identified by the official CMS online tool (https://icd10cmtool.cdc.gov/?fy=FY2024), by using the keyword “pressure ulcer”. The codes were validated by the pressure ulcer codes list from icd10Data.com. A total of 149 pressure ulcer codes were identified, representing 25 different anatomical sites. There are different codes for left and right anatomical sites (e.g., left elbow vs right elbow PUs are represented by different codes).

Reviewer comment: The use of “worst stage” and “multisite” constructs appear reasonable but requires validation. How was overlap between left and right sides managed?

Authors response: We decided to study left and right site pressure ulcer codes as different anatomical sites and locations. We made this decision since there are several implications that opposite pressure ulcer sites may carry implications which are not random but may have to do with different conditions or treatment/management requirements. Here is the text that we added on the updated manuscript.

There are different codes for left and right anatomical sites (e.g., left elbow vs right elbow PUs are represented by different codes). PUs on patients who had PU on both sites of the same location were treated as different PUs, and not merged. We made this decision because of different implications that opposite sites (left vs right) may be holding such as underlying clinical and functional factors, instead of just being random occurrence.”

Reviewer comment: Adjusting only for age, sex, primary diagnosis, and transfer source seems insufficient; including comorbidity measures (e.g., Charlson Comorbidity Index) would strengthen the model.

Authors response: We actually discussed about this with my colleague when conducted the study. We ended up deciding not to include comorbidities as covariates, because they are often interrelated with both ulcer development and outcomes. This could have “obscured” these relationships. Basically, comorbidities may function as mediators rather than just confounders reflecting patient frailty is “between pressure ulcer severity and mortality/length of stay. We did decide to control for the principal diagnosis though, because this knowledge functions as “adjustment” for the clinical context in which PUs occurred (i.e., telling surgical from medical cases). We added the above text in the methods section of the updated manuscript as well.

Reviewer comment: Finally, I recommend specifying the statistical software, significance thresholds, and confidence intervals consistently.

Authors response: The statistical analysis section (2.2) of the paper ends with the following statement:

“Statistical analyses were performed using IBM SPSS Statistics for Windows, Version 29.0 (IBM Corp., Armonk, NY). A two-tailed alpha level of 0.05 was used to determine statistical significance.”

Reviewer comment: In the Results, the text frequently repeats data already presented in the tables. I suggest you condense descriptions and highlight key patterns only.

Authors response: Thank you. This was an omission. We updated all the sections of the Results where we had originally repeated specific statistics that are already included in the tables. Now there are only references to high key patterns only, or summaries of the direction of findings, and not specific statistical results and p-values. Please visit the track-edit version of the updated manuscript to review all changes in the Results section.

Reviewer comment: The PULS formula is useful, but rationale for weighting by stage severity should be justified and referenced. At present, no PULS index is recognized in the literature for assessing the severity of pressure ulcers based on their anatomical location.

Authors response: The PULS formula is not a previous formula. This is simply an acronym that we gave to the “Pressure Ulcer Locality Stage” score which we calculated from our data and introduced in this paper. PULS is basically the location-specific stage distribution across all our patient population. It does not assess anything. It is simply a descriptive metric measuring the “tendency” of each anatomical site to have higher or lower stage pressure ulcers. It roughly represents the stage distribution of pressure ulcers for each anatomical site.

Reviewer comment: The Discussion section requires a clearer structure: it should begin with the key findings, followed by their interpretation, then address strengths and limitations, and conclude with the implications. In its current form, the discussion blends speculative reasoning with the results. It is also important to avoid redundancy, as several sentences repeat the same idea. Finally, additional references should be included to support the claims made in this section.

Authors response: We decided to reconstruct and add to the discussion significantly. The discussion has a very different structure now, to add clarity. The new flow is exactly as you recommended. While we were rewriting, we were removing repeated ideas, and merged other ideas in a meaningful way. We also added three more references. Please review the resubmitted “tracked-changes” manuscript to review the updated Discussion section.

Reviewer comment: The conclusions are somewhat repetitive and should be refined in light of the discussion.

Authors response: We completely re-wrote the Conclusions to be more directly associated with the discussion and less generic. Below is the new Conclusions section.

The overall PU prevalence of 3.7% aligns with national trends, confirming that PUs remain a persistent issue in hospital settings. Sacral, buttock, and heel ulcers were the most frequent PUs. Stage 2 ulcers predominated. This aligns with the tendency for early tissue breakdown starting to become clinically detectable at stage 2, rather than Stage 1. The numerous unstageable ulcers, particularly in the heels and head, shows ongoing challenges in documentation and staging accuracy. These findings suggest that training, resource availability, and assessment protocols can improve classification and detection of PUs.

PU site and multiplicity were stronger predictors of LOS and mortality than stage alone Length of stay. Sacral, hip, and head region PUs were independently associated with longer LOS and higher mortality, likely reflecting greater systemic frailty and comorbid burden. The presence of multiple ulcers amplified these risks, supporting the interpretation that extensive ulceration signals a global decline in patient resilience.

Interestingly, higher ulcer stage was not directly linked to increased mortality. This may be because of underlying illness severity or institutional coding and care practices that affect outcome patterns.

Overall, authors believe that PUs should be viewed both as indicators and contributors to negative hospital outcomes. The feedback loop between prolonged LOS and PU severity highlights the importance of early prevention, monitoring, and standardization and consistency in reporting. While the study was not designed to examine associations of PUs with comorbidities, future research should also explore hospital-level determinants and patient-level comorbidities to better identify high-risk patient subgroups.

We would like to emphasize the need for patient safety protocols that prioritize prevention but also early detection. Validated risk assessments, pressure redistribution intervention, and staff education are foundational to controlling PU progression. Training should focus on early recognition of initial and oftentimes hard to recognize skin changes. These preventive measures will also lessen the effects of PUs on hospital resource use and system-level outcomes.

Reviewer 2 Report

Comments and Suggestions for Authors

Dear Editor, Dear Authors,

The article is highly informative, well written, and provides valuable informations into the management and statistical outcomes of pressure ulcers, including their relationship with mortality and LOS. Notably, your findings suggest that higher-stage pressure ulcers were not directly associated with greater 360-day mortality risk, an observation of considerable interest and great novlty.

A few minor points for refinement:

  1. Please define all abbreviations at first mention (e.g., “pressure ulcer (PU)” in abstract). 

  2. In Table 2, do the outlined cells have a specific meaning, or is this a formatting artifact? Please clarify or standardize the formatting.

  3. For the statistical variables, include ranges, standard deviations, and confidence intervals where applicable to enhance interpretability.

  4. In the Discussion, add a more practical angle: briefly outline how these findings could influence PU prevention strategies, bedside management, and treatment pathways (e.g., risk stratification, resource allocation, and follow-up priorities).

Thanks for attention

Author Response

Thank you for your review. After an extensive meeting with my co-author, we read your critique and are explaining here how we addressed your questions. In green is your comment and dark purple our response. We also included in the resubmission portal of MDPI, a “track-changes” version of the manuscript, so that you can see the changes we made.

Reviewer comment: Please define all abbreviations at first mention (e.g., “pressure ulcer (PU)” in abstract).

Authors response: We reviewed the document and added abbreviations at first mention where missing. Thank you.

Reviewer comment: In Table 2, do the outlined cells have a specific meaning, or is this a formatting artifact? Please clarify or standardize the formatting.

Authors response: The outlined cells represent the most frequent stages per pressure ulcer anatomical site. Reading your comment, we thought that you are right. They are confusing and they do look like a formatting error. We removed them and used an asterisk for those cell values, and a table footnote explaining that these are the most frequent stages per pressure ulcer anatomical site.

Reviewer comment: For the statistical variables, include ranges, standard deviations, and confidence intervals where applicable to enhance interpretability.

Authors response: We added confidence intervals and now are present in both regression analyses.

Reviewer comment: In the Discussion, add a more practical angle: briefly outline how these findings could influence PU prevention strategies, bedside management, and treatment pathways (e.g., risk stratification, resource allocation, and follow-up priorities).

Authors response: We reconstructed and add to the discussion significantly, to address other reviewers’ remarks. The discussion has a different structure to add clarity. recommended. While we were rewriting, we were removing repeated ideas, and merged other ideas in a meaningful way. We also added three more references. To address your specific comment, we added a small paragraph and citations to add a more practical angle to the discussion. This is the added text:

These findings should trigger discussions to improve bedside PU prevention and management. Risk stratification can be refined. We also recognize the need for better allocation for support devices as well as specific protocols to patients with ulcers in high-risk locations (sacral, hip, head, upper back) or multiple PUs. Bedside, this translates to prioritized skin assessments for these patients. Treatment pathways should also be considered to be adjusted, to initiate earlier wound care for PUs at high-risk sites.

Reviewer 3 Report

Comments and Suggestions for Authors

I reviewed the manuscript with the title “Beyond Staging: The Role of Pressure Ulcer Site and Multiplicity in Hospital Mortality and Length of Stay”

It is an interesting study and will help towards improving quality of life. Some of the suggestions below.

Abstract: of the study is well written.

Introduction:  is well written. It is unclear when authors discuss about mortality rate- is it associated with PU only? Or mortality due to health conditions and also have PUs of various stages?

Materials and methods: well described. Will be useful to know the duration of data collection, if collected at various hospital sites (although Medicare patient data) highlighting if same or different health care practices at various hospital sites.

Results: good description with good number of patients selected for the study.

Discussion:

Lines 311-320- authors are restating the results obtained

Except for first paragraph there are only 2 references cited. Discussion should mainly contain the results obtain in the study and comparison to that of known scientific knowledge which is lacking in the current draft of discussion.

Conclusion: in line with the study and observations

References: look fine

Overall, I think it is interesting work. It is a well written manuscript. In terms of rationale of the study, authors have tried to have some correlation with mortality rate. From the findings and description in this manuscript it is not quite clear if PU are related/cause for mortality (which I don’t think they are). It can be looked at as co-morbidity for prolonged hospitalisation/LOS and authors have presented the correlation with length of stay and stages of PU. It will also be useful to have some more details on duration of the data collected (although retrospective), if specific hospital or various hospitals (how many?) etc. details. Discussion should be focused and in line with current scientific understanding.

Author Response

Reviewer 3

Thank you for your review. After an extensive meeting with my co-author, we read your critique and are explaining here how we addressed your questions. We also included in the resubmission portal of MDPI, a “track-changes” version of the manuscript, so that you can see the changes we made.

Reviewer comment: Introduction is well written. It is unclear when authors discuss about mortality rate- is it associated with PU only? Or mortality due to health conditions and also have PUs of various stages?

Authors response: Thank you for the comment. The mortality rate discussed in our study refers to all-cause mortality among hospitalized patients who also had pressure ulcers of various stages. We have clarified this in the revised manuscript to indicate that mortality was not limited to deaths directly caused by pressure ulcers.

Reviewer comment: Materials and methods are well described. Will be useful to know the duration of data collection, if collected at various hospital sites (although Medicare patient data) highlighting if same or different health care practices at various hospital sites. 

Authors response: Thank you for the comment. We added to the manuscript the following paragraph to address your comment:

“Inpatient dataset is a de-identified claims dataset that includes information on inpatient hospital stays for Medicare patients for the year 2019. The dataset includes every Medicare inpatient admission that happened in hospitals located in the United States during that year. The dataset is made available directly via CMS. The researchers did not extract the original data themselves. Instead, the extracted and cleaned data is made available directly through CMS. CMS themselves compiles the dataset and releases them on an annual and quarterly basis.”

Reviewer comment: Lines 311-320- authors are restating the results obtained

Authors response: We rewrote the section so that it does not restate the results.

Reviewer comment: Except for first paragraph there are only 2 references cited. Discussion should mainly contain the results obtain in the study and comparison to that of known scientific knowledge which is lacking in the current draft of discussion. 

Authors response: We added several new citations to the discussion section which is directly related to the current knowledge about pressure ulcer risk-factors and implications.

Reviewer comment: Overall, I think it is interesting work. It is a well written manuscript. In terms of rationale of the study, authors have tried to have some correlation with mortality rate. From the findings and description in this manuscript it is not quite clear if PU are related/cause for mortality (which I don’t think they are). It can be looked at as co-morbidity for prolonged hospitalization/LOS and authors have presented the correlation with length of stay and stages of PU.

Authors response: We agree that pressure ulcers are unlikely to be a direct cause of mortality. In our study, we examined mortality as an outcome associated with the presence and stage of PUs, acknowledging that PUs may serve as indicators of overall patient frailty or severity of illness rather than direct causes of death. The updated discussion section emphasizes the role of PUs as comorbid conditions contributing to mortality and prolonged hospital stays.

Reviewer comment: It will also be useful to have some more details on duration of the data collected (although retrospective), if specific hospital or various hospitals (how many?) etc. details. Discussion should be focused and in line with current scientific understanding.

Authors response: We added this information, that the data are from all Medicare patients from the entire United States. We also added more details about the data collection (1 full year of data for 2019).

Round 2

Reviewer 1 Report

Comments and Suggestions for Authors

Dear authors,
thank you for submitting the revised version of your manuscript.
The quality of the manuscript has been improved and I am pleased to recommend it for acceptance.

Author Response

Thank you for giving us a great opportunity to improve the article in the first round. I appreciate on behalf of my co-author your feedback.

Reviewer 3 Report

Comments and Suggestions for Authors

I read the revised version of the manuscript.

It's looking better now.

Regarding conclusion, I suggest to make it concise and only conclusion of the study. 

Line 754: change "authors believe" 

Author Response

Thank you for reviewing the article for the second round. We removed the first 3 lines of the conclusion because we agree with your comment that it repeats Discussion information. We made some other adjustments to the Conclusion section to make it more concise per your recommendation, and also addressed your wording comment.

Thank you for your timely feedback!